# OpenReview forum: "ReflFlow: Learning Geometry-Guided Ray Tracing for Dynamic Specular Reconstruction"
_ICML.cc/2026/Conference — ICML 2026 regular_

### Official Review · Reviewer_22GR · 2026-03-03

**Soundness:** 3
**Presentation:** 3
**Significance:** 2
**Originality:** 2
**Overall Recommendation:** 3
**Confidence:** 4

**Summary:**

The paper tackles the problem of view synthesis of specular objects in dynamic scenes. This is a challenge in novel view synthesis because view-dependent effects like specularities can be often confused with object motion. Furthermore, popular 3D reconstruction techniques like 3DGS or 2DGS often produce inaccurate surface normals, which are important for modeling reflections. To address this problem, the paper proposes a new differentiable rendering technique which explicitly models view-dependent effects with ray tracing. The authors state three technical contributions: (1) a representation for time-varying Gaussians, (2) separate renderers for diffuse and specular components, and (3) a coarse-to-fine training strategy.

**Compliance With Llm Reviewing Policy:**

Affirmed.

**Final Justification:**

During the rebuttal, the authors included more results to highlight the quality of the reconstructed normals.

However, a major point is not addressed, which is that there is no way to guarantee that specularities are not being modeled falsely as object motion. Due to this omission, I would like to maintain my score.

Nonetheless, I am not opposed to acceptance either. As noted by the authors, this is a shared limitation of many works in view synthesis. I myself have not seen a compelling solution to the specularity / motion disentanglement problem yet.

**Key Questions For Authors:**

1. Referring back to the weaknesses, what exactly does “dynamic” mean in this paper? All the scenes rendered seem to be of static objects. Furthermore, how can you ensure that the model is not “cheating” by representing view-dependent effects falsely as motion of Gaussians?
2. From Figure 2, I have a hard time seeing the difference between the diffuse and final images. Could the authors please include more examples of the decomposition between diffuse and specular components?
3. Could the authors also render a video of the depth and normal maps? This would help demonstrate that the geometry is view consistent, free of artifacts, and free of aliasing.
4. What is the main difference between this work and Xie et al., 2024, which proposed the environment Gaussian representation, and this work?

**Limitations:**

Yes

**Strengths And Weaknesses:**

## Strengths
- The problem statement is clearly stated. The method is technically sound and seems to be complete. The experiments and ablation studies help demonstrate the benefits of the method over prior work.
- The included video demonstrates how the generated reconstructions faithfully reproduce view-dependent effects, while remaining 3D consistent.
- The produced normal maps are high quality compared to prior work, mainly due to the alignment of the normals with normals predicted from a monocular normal network.

## Weaknesses

The main weakness I see in this paper arises from confusion of what the term “dynamic” means in this paper. Typically in novel view synthesis, dynamic reconstruction means reconstructing both the 3D geometry of the scene, and the motion of objects in the scene (see [A] for an example). However, the paper presented only shows renders of static scenes, albeit with specular reflections. I inspected the NeRF-DS dataset used and did not see any object motion. So what exactly does “dynamic” mean in this context?

The fact that the scenes used have no object motion raise confusion on what exactly the residual  $\Delta \mathbf{G}^t$ is doing. If the scene is static, shouldn’t there be no change in Gaussian parameters across time? This aspect of the method does not seem physically-plausible. Yet in practice, Table 3 shows that using the residual significantly improves reconstruction quality by over 2.5 dB. This inconsistency between the method and empirical results makes it seem that the model may be falsely representing specularities as residuals in the diffuse Gaussian parameters, which is not physically accurate.

To address this inconsistency, I suggest that the authors show results on scenes with both object motion and specularities, or remove the time-varying residual and re-run results on the NeRF-DS dataset. I notice that quantitative results are shown on the HyperNeRF dataset, which is dynamic, but no visual results are shown. This is the main problem I see justifying my score, and I hope the authors clarify this use of the term "dynamic" in the rebuttal or a revised paper.

In addition, there are some other minor weaknesses with the paper:
- The technical contributions seem limited. As stated in the related work section, many prior works have already proposed techniques for rendering time-varying Gaussians. Furthermore, the environment Gaussian technique is already mostly described in Xie et al., 2024. The authors should clarify the technical differences between the proposed method and prior work.
- This does not affect my rating, but I think the paper would be of limited interest to the ICML community. The paper is mostly focused on the graphics application, which would seem more appropriate for CVPR or SIGGRAPH.


[A] Li et al., DynIBaR: Neural Dynamic Image-Based Rendering. CVPR 2023

---

> ### Author Rebuttal · Authors · 2026-03-31
>
> We thank Reviewer 22GR for the careful review and insightful questions. We address each concern below.
>
> **Clarification: NeRF-DS Scenes Contain Real Object Motion (Main Weakness & Q1)**
>
> We appreciate the reviewer's careful inspection. We respectfully clarify that NeRF-DS (Yan et al., 2023) scenes do contain real object motion. The original paper explicitly describes them as "real-world dynamic scenes with moving or deforming specular objects." Specifically: Plate contains a hand continuously rotates a metallic plate; As contains a hand squeezes and deforms aluminum foil (non-rigid); Sieve contains a hand moves a metallic sieve; Bell contains a hand strikes a bell; Cup/Press/Basin contains hands interact with objects causing displacement or deformation.
>
> The motion may appear subtle between adjacent frames in our figures, but across the full sequence (50-200 frames), the geometric changes are substantial. We refer the reviewer to the original NeRF-DS paper and its supplementary video, which clearly demonstrate these object motions.
>
> Regarding the role of $\Delta \mathcal{G}^t$: it captures real geometric motion of objects (hand displacement, object rotation/deformation) and small material adjustments due to monocular ambiguity. The +2.7 dB improvement from the deformation network (Table 3) directly reflects modeling this real motion. Without it, ghosting from unmodeled object displacement severely degrades image quality. Table 3 averages over four scenes (As, Press, Plate, Bell) with diverse geometry and motion, and the deformation network shows consistent gain across this range. If $\Delta \mathcal{G}^t$ were merely absorbing view-dependent effects, such consistent improvement would not occur.
>
> **Q2: Diffuse/Specular Decomposition Visualization**
>
> We provide a video showing the diffuse component, specular component, and final rendering separately for the Plate scene: https://drive.google.com/file/d/1mi3TQ_rdkFx45PHtLK3eZsA07FIF9N0S/view?usp=drive_link (diffuse) and https://drive.google.com/file/d/12LwxjiCSLq0q3akC6uCv_khlOZSE4WXd/view?usp=drive_link (specular). The specular branch captures distinct reflection patterns on the metallic surface (moving highlights, environment reflections) that are absent from the diffuse rendering. The difference is more pronounced in highly specular regions. In regions with weak specularity, the diffuse and final images naturally appear similar, which is expected behavior, not a failure of decomposition.
>
> **Q3: Depth and Normal Map Rendering**
>
> We provide rendered normal maps for the Plate scene across multiple viewpoints, demonstrating view-consistent, smooth, and alias-free geometry: https://drive.google.com/file/d/1z_U81EjLA1Q7FMf8pjEyzeJUA3zvR67e/view?usp=drive_link. The temporal consistency of our normal maps benefits from NormalCrafter supervision (see our response to Reviewer EkiH, Q1 for a sensitivity analysis on this choice).
>
> **Q4: Difference from Xie et al., 2024 (EnvGS)**
>
> The fundamental difference is that EnvGS is a static method. It has no deformation network, no time-conditioned components, and cannot handle dynamic scenes. In Table 1, EnvGS is evaluated as-is following SpectroMotion's baseline protocol, which explains its poor performance on dynamic data (20.21 mean PSNR vs our 24.91). Our method extends the environment Gaussian idea to the dynamic setting by introducing time-conditioned residual corrections ($F_{\theta_{env}}$) and coupling it with Material-Augmented 2DGS for geometry-accurate ray tracing, both of which are absent in EnvGS.
>
> **On Technical Novelty**
>
> The core contribution is not a simple integration. Dynamic specular reconstruction from monocular video requires these components to interact in a tightly coupled manner. (1) Material-Augmented 2DGS is specifically designed for reflection-aware ray tracing. Standard 2DGS does not support view-dependent material properties (specular tint), which are prerequisites for physically meaningful reflection computation. (2) Dynamic Environment Gaussians with time-conditioned residuals are fundamentally different from static EnvGS and essential for dynamic scenes. (3) The hybrid rasterization-ray tracing pipeline is made feasible only through our reflection-oriented parameterization and coarse-to-fine strategy. Without coarse-to-fine, PSNR drops from 21.10 to 15.28 on Plate (-5.8 dB). Table 3 confirms tight coupling: each component provides significant incremental gain (F_θG +2.7 dB, F_θenv +1.3 dB, L_norm +1.9 dB, L_tc-norm +3.1 dB).

---

> > ### Author Rebuttal · Reviewer_22GR · 2026-04-01
> >
> > Thank you for taking the time to respond to my questions. The new video results will make the paper stronger. The normals in particular look very high quality and consistent.
> >
> > However, a major point is not addressed, which is that there is no way to guarantee that specularities are not being modeled falsely as object motion. I think a specific ablation on this(with accompanying videos) would make the paper's claim of being physically-based significantly stronger. Results on scenes with more dynamic movement than just incidental motion would also be important to support the paper's claims of working on dynamic scenes.
> >
> > I will update my final review after discussing with the other reviewers.

---

> > > ### Author Response · Authors · 2026-04-03
> > >
> > > Thank you for reviewing the new videos and for the positive feedback on our normal quality. We address your remaining concerns below.
> > >
> > > **1. Disentanglement of Specularities and Object Motion**
> > >
> > > We acknowledge that in a monocular setting, perfect disentanglement between geometric motion and view-dependent appearance is inherently challenging. Some degree of ambiguity is shared by all monocular dynamic methods (including SpectroMotion, 4DGS, Deformable 3DGS), as no method can fully resolve this without multi-view supervision. This is precisely why we revised our claim from "physically based" to "physically informed" (see our response to Reviewer mYKQ W1).
> > >
> > > That said, our design actively mitigates this ambiguity, and we present structural and empirical evidence:
> > >
> > > (1) **No-RT ablation:** We provide a side-by-side video comparing our full model with a variant where the ray tracing branch is removed (w/o RT): https://drive.google.com/file/d/1bYJryg3jv3eAQtzI8VtYJJ9DihEOxhE8/view?usp=drive_link. Without ray tracing, specular quality degrades visibly: reflections become blurry and lose fine detail. This confirms that specular information is predominantly carried by the ray tracing pipeline, not by $\Delta \mathcal{G}^t$.
> > >
> > > (2) **Distinct failure modes (Table 3):** Removing the deformation network causes ghosting (geometric artifacts from unmodeled motion), while removing Dynamic Environment Gaussian primitives causes flat/missing reflections (specular artifacts). These qualitatively different failure modes indicate the two branches carry largely distinct information.
> > >
> > > (3) **Coarse-to-fine training:** Phase 1 trains only the diffuse branch (including deformation), grounding $\Delta \mathcal{G}^t$ in geometric reconstruction before the specular branch is activated. This structural ordering reduces early-stage leakage of specular effects into deformation.
> > >
> > > **2. Scenes with Larger Dynamic Movement**
> > >
> > > Regarding scenes with larger motion: to our knowledge, NeRF-DS is currently the only benchmark for dynamic specular reconstruction, and all comparable methods (such as SpectroMotion) evaluate on the same dataset. We agree that a benchmark with larger object motion would be valuable and consider constructing one as future work.

---

### Official Review · Reviewer_nhMk · 2026-03-10

**Soundness:** 3
**Presentation:** 3
**Significance:** 2
**Originality:** 2
**Overall Recommendation:** 4
**Confidence:** 3

**Summary:**

This paper proposes ReflFlow, which combines Residual Material-Augmented 2D Gaussian Splatting with a Dynamic Environment Gaussian representation and a hybrid rasterization–ray tracing pipeline. Experimental results demonstrate improved quantitative metrics and sharper specular reconstruction compared to previous approaches.

**Compliance With Llm Reviewing Policy:**

Affirmed.

**Final Justification:**

The authors' rebuttal has addressed my concerns, and I tend to increase my previous ratings. Overall, I would recommend **weak accept**.

However, since I am not an expert in this area, please consider weighting my reviews and ratings appropriately.

**Key Questions For Authors:**

1. How sensitive is the performance to the quality of the external normal estimator? What happens if this supervision is removed?

2. Since the diffuse/specular decomposition is weakly supervised, are there cases where the decomposition becomes degenerate?

3. How does the proposed Dynamic Environment Gaussian compare with other environment representations in terms of memory and accuracy?

**Limitations:**

No.
Please refer to weaknesses and key questions.

**Strengths And Weaknesses:**

Strength:

This paper identifies two key limitations of current dynamic specular reconstruction methods: inaccurate normal estimation in 3DGS and the limited expressiveness of environment maps. Besides, explicit diffuse/specular decomposition and reflection-based rendering make the method more physically interpretable than purely view-dependent appearance modeling. The included ablation experiments also validate the contributions of the time-conditioned residual network, residual correction network and geometry-aligned normal loss.

Weaknesses:

(1) Many components appear incremental, which have already existed in the literature. The authors mainly combine them rather than introducing fundamentally new ideas.

(2) This paper introduces NormalCrafter for temporal consistency supervision, which raises the concern about the potential bias.

(3) Since the proposed method introduces ray tracing and additional Gaussian representations, the computational overhead and rendering speed compared to real-time 3DGS pipelines and other baselines are not clearly reported (except comparison with SpectroMotion as shown in appendix).

(4) Some components lack implementation details.

---

> ### Author Rebuttal · Authors · 2026-03-31
>
> We thank Reviewer nhMk for the constructive feedback. We address each weakness and question below.
>
> **W1: Incrementality**
>
> The core contribution is not a simple integration. Dynamic specular reconstruction from monocular video requires these components to interact in a tightly coupled manner, and prior methods have not provided a viable formulation. Specifically:
>
> - Material-Augmented 2DGS is specifically designed for reflection-aware ray tracing: standard 2DGS does not provide stable normals or support view-dependent material properties (specular tint), which are prerequisites for physically meaningful reflection computation.
> - Dynamic Environment Gaussian is fundamentally different from the static EnvGS formulation. Our time-conditioned residual correction is essential for handling illumination changes in dynamic scenes, a capability static EnvGS does not provide.
> - The hybrid rasterization-ray tracing pipeline is made feasible only through our reflection-oriented parameterization of 2D Gaussians and coarse-to-fine training. Without this coupling, optimization is unstable (removing coarse-to-fine: 15.28 vs 21.10 PSNR on Plate, -5.8 dB).
>
> Table 3 confirms tight coupling: each component provides significant incremental gain (F_θG +2.7 dB, F_θenv +1.3 dB, L_norm +1.9 dB, L_tc-norm +3.1 dB). These are interdependent components, not interchangeable modules.
>
> **W2: NormalCrafter Potential Bias**
>
> Please see our response to Reviewer EkiH Q1. We replaced NormalCrafter with StableNormal (SIGGRAPH Asia 2024, per-frame diffusion-based). Result: training collapsed (loss=NaN) due to severe inter-frame normal flickering. This demonstrates that temporal consistency is the critical requirement. Importantly, even without $\mathcal{L}_{tc\text{-}norm}$, our method still achieves ~20.7 PSNR on Plate (stable training), confirming the framework is not dependent on external normal priors.
>
> **W3: Computational Overhead**
>
> We provide an expanded efficiency comparison:
>
> | Method                   | Training  | FPS    | Task                  |
> | ------------------------ | --------- | ------ | --------------------- |
> | 4DGS (CVPR'24)           | ~30 min   | 82     | Dynamic (no specular) |
> | GaussianShader (CVPR'24) | ~35 min   | ~100   | Static specular       |
> | SpectroMotion (CVPR'25)  | 1.1 h     | 33     | Dynamic specular      |
> | **ReflFlow (Ours)**      | **2.8 h** | **32** | **Dynamic specular**  |
>
> 4DGS and GaussianShader address simpler tasks (no specular or no dynamics). Among methods handling both, our FPS matches SpectroMotion (32 vs 33). The longer training stems from more iterations (60k vs 40k) for the three-phase schedule, not higher per-step cost. We consider this acceptable for +0.74 dB PSNR improvement (Table 1).
>
> **W4: Implementation Details**
>
> We provide key details: Deformation network: 8-layer MLP, 256 hidden units, with Fourier-embedded 3D coordinates and time input. Initialization: dataset-provided points.npy (no SfM dependency). Specular tint $s_{tint}$: scalar value (not 3-channel RGB), acting as Fresnel-style intensity modulation. Standard 3DGS densification and pruning are applied. We will add a complete implementation details section in the revision.
>
> **Q1: Sensitivity to Normal Estimator**
>
> Addressed in W2 above and our response to Reviewer EkiH Q1.
>
> **Q2: Degenerate Decomposition**
>
> We provide a decomposition visualization for the Plate scene: https://drive.google.com/file/d/1mi3TQ_rdkFx45PHtLK3eZsA07FIF9N0S/view?usp=drive_link (diffuse) and https://drive.google.com/file/d/12LwxjiCSLq0q3akC6uCv_khlOZSE4WXd/view?usp=drive_link (specular). In highly specular regions (metallic surface), the specular branch captures clear reflection patterns. The coarse-to-fine training (Phase 1: diffuse only, Phase 2: specular only, Phase 3: joint) structurally prevents global degeneracy, as each branch is independently grounded before joint optimization. While this strategy successfully prevents one branch from absorbing everything (as shown in the provided videos), localized ambiguity can occasionally occur in highly untextured, mirror-like regions where diffuse color is entirely overwhelmed by environment reflections, making the separation ill-posed without multi-view cues. However, this is a fundamental limitation of the monocular setup rather than a failure of the decomposition mechanism.
>
> **Q3: Comparison with Other Environment Representations**
>
> Please see our response to Reviewer EkiH Q3 for a detailed comparison. In summary: Table 1 baselines serve as proxies for alternative representations (SpectroMotion = dynamic env map, 24.17; EnvGS = static env Gaussians, 20.21), and our Dynamic Env Gaussians (24.91) outperform both. Table 3 isolates the contribution at +1.28 dB.

---

> > ### Author Rebuttal · Reviewer_nhMk · 2026-04-02
> >
> > Thanks for the response. While I still have some hesitation about the novelty and originality of this work, the authors' rebuttal has addressed my concerns, and I will raise my score to 4.
> >
> > However, since I am not an expert in this area, please consider weighting my reviews and ratings appropriately.

---

### Official Review · Reviewer_mYKQ · 2026-03-11

**Soundness:** 3
**Presentation:** 3
**Significance:** 3
**Originality:** 3
**Overall Recommendation:** 4
**Confidence:** 2

**Summary:**

The paper proposes ReflFlow, a dynamic specular reconstruction framework that combines a Residual Material-Augmented 2D Gaussian Splatting (2DGS) representation with a Dynamic Environment Gaussian field and a hybrid rendering pipeline (rasterization for diffuse, ray tracing for specular). By leveraging exact normals from 2DGS and learning a time-varying environment representation, the method aims to compute reflection directions accurately and better capture near-field, high-frequency specular details. A coarse-to-fine training schedule stabilizes the inherently ill-posed diffuse/specular decomposition. Experiments on NeRF-DS and HyperNeRF report state-of-the-art or competitive performance with sharper, temporally consistent reflections.

**Compliance With Llm Reviewing Policy:**

Affirmed.

**Key Questions For Authors:**

1. How sensitive is the method to the quality of NormalCrafter normals? Do you have results replacing it with other priors or ablating Ltc-norm across different scenes?
2. How are 2DGS rotations (t_u, t_v) parameterized dynamically (i.e., what is r)? Please clarify how Δr^t is applied to update tangents and ensure orthogonality and stability.
3. How do you handle occlusions along reflection rays and avoid multi-bounce artifacts (e.g., specular self-reflections)? Is there an explicit occlusion test or transmittance-based stopping criterion?
4. Can you quantify near-field reflection improvements (e.g., an object reflecting another nearby object) on a controlled synthetic benchmark where ground-truth reflection geometry and directions are known?
5. What are the memory and runtime overheads introduced by the environment Gaussians across datasets? Please include environment Gaussian counts and scene-scale variability.
6. Could you clarify the corrupted term in Eq. (14) and provide the precise transmittance definition (T_i) used in practice?

**Limitations:**

yes

**Strengths And Weaknesses:**

# Strengths:
- The integration of 2DGS to obtain exact surface normals for reflection-ray computation is a compelling design choice that directly addresses normal-estimation errors in 3DGS-based pipelines.
- Dynamic Environment Gaussians extend environment-map concepts, enabling near-field, higher-frequency specular modeling beyond low-res far-field maps.
- The hybrid rasterization + ray tracing pipeline, combined with a material-aware specular tint parameter and a staged training schedule, is a coherent system-level contribution.

# Weaknesses:
- The “physically based” claim is overstated: the specular branch lacks a clear microfacet BRDF (e.g., roughness, Fresnel, energy conservation) and instead relies on a learned αspec blending from specular tint, which is heuristic and may not be physically accurate.
- Potential leakage/entanglement between main-scene Gaussians and Dynamic Environment Gaussians is not rigorously controlled; no explicit spatial/semantic separation, masking, or consistency constraints are described.
- Missing analysis on how near-field vs far-field reflections are handled and validated; no synthetic benchmark with ground-truth reflection directions or known environment geometry to quantitatively evaluate reflection accuracy.

---

> ### Author Rebuttal · Authors · 2026-03-31
>
> We are grateful to Reviewer mYKQ for the encouraging assessment and the insightful technical questions. We address each point below.
>
> **W1: "Physically Based" Claim**
>
> We agree and thank the reviewer for this observation. We will revise "physically based" to "physically informed" throughout the paper. Our specular branch does not implement a full microfacet BRDF (no roughness, Fresnel, or energy conservation). Instead, it follows the physically motivated principle of separating diffuse and specular components and computing reflection directions from accurate surface normals via explicit ray tracing. This is consistent with all prior monocular specular methods (EnvGS, SpecNeRF, Ref-NeRF, SpectroMotion), which also adopt single-bounce mirror reflection, the only specular component that can be reliably identified from monocular video. Complex effects (glossy BRDFs, microfacets, multi-bounce interreflections) are fundamentally unobservable under monocular constraints.
>
> **Q1: NormalCrafter Sensitivity**
>
> Please see our response to Reviewer EkiH Q1 for detailed experimental results, including the StableNormal replacement experiment (training collapse due to temporal inconsistency) and the ablation without $\mathcal{L}_{tc\text{-}norm}$ (~20.7 PSNR on Plate, stable training). The key finding is that temporal consistency, not a specific estimator, is the critical requirement, and the framework remains sound without external normal priors.
>
> **Q2: 2DGS Rotation Parameterization**
>
> Our 2DGS parameterizes orientation via a unit quaternion $q = (w,x,y,z)$, from which the rotation matrix $R = [t_u, t_v, t_w]$ is constructed. The surface normal $t_w$ is the third column of $R$. Our deformation network predicts a 4D quaternion residual $\Delta q^t$, applied as $q' = q + \epsilon \cdot \Delta q^t$ with a small scaling factor, where normalization is handled internally by the rasterizer. This small-residual additive design in quaternion space is simple and stable for the small per-frame deformations in our setting. The well-defined normal from 2DGS is a key advantage over 3DGS (which requires approximation), enabling accurate reflection ray computation (Eq. 4) essential for our physically informed specular modeling.
>
> **Q3: Occlusion Handling Along Reflection Rays**
>
> We trace each reflection ray through the environment Gaussian field using alpha compositing with accumulated transmittance $T_i = \prod_{j=1}^{i-1}(1-\alpha_j)$. Ray marching terminates when transmittance falls below a threshold ($T_i < \epsilon$), providing an implicit occlusion test. We currently model single-bounce reflections only; multi-bounce inter-reflections are explicitly noted as a limitation (Sec. 6). This single-bounce assumption is shared by all comparable methods (EnvGS, SpecNeRF, Ref-NeRF, SpectroMotion).
>
> **Q4: Near-Field Reflection Quantification**
>
> We do not currently have a synthetic benchmark with ground-truth reflection geometry. However, qualitative evidence of near-field modeling is visible in our results: the Press scene shows reflections of nearby objects (apples and bread on the table) on the metallic surface, which environment maps cannot capture due to the far-field assumption. Table 3 confirms that the Dynamic Environment Gaussian (which models near-field) improves PSNR by 1.28 dB over ablated versions. We agree a synthetic evaluation would strengthen the paper and plan to include one in the revision.
>
> **Q5: Memory and Runtime Overhead**
>
> | Method              | Gaussians                       | GPU Memory   | FPS    | Training  |
> | ------------------- | ------------------------------- | ------------ | ------ | --------- |
> | **ReflFlow (Ours)** | **~55K** (scene ~46K + env ~9K) | **1,974 MB** | **32** | **2.8 h** |
> | SpectroMotion       | ~50K                            | ~1,800 MB    | 33     | 1.1 h     |
>
> Environment Gaussians add ~16% to the Gaussian count with minimal overhead. The rendering FPS is comparable to SpectroMotion (32 vs 33) since ray tracing only operates on the lightweight environment Gaussian set. The additional training time (2.8h vs 1.1h) stems from the three-phase coarse-to-fine schedule and ray tracing, yielding +0.74 dB mean PSNR improvement over SpectroMotion (Table 1).
>
> **Q6: Eq. (14) Clarification**
>
> We apologize for the confusion. $T_i$ denotes accumulated transmittance along the reflection ray: $T_i = \prod_{j=1}^{i-1}(1-\alpha_j)$, where $\alpha_j$ is the opacity of the $j$-th environment Gaussian intersected along the ray. The specular color is then $C_s = \sum_i T_i \alpha_i c_i$, following standard alpha compositing. We will clarify this in the revision.
>
> We appreciate Reviewer mYKQ's positive assessment and believe the StableNormal experiment, "physically informed" correction, and technical clarifications above fully address the raised concerns. We will incorporate all suggested revisions.

---

> > ### Author Rebuttal · Reviewer_mYKQ · 2026-04-05
> >
> > Thank you for your rebuttal and for addressing my concerns. I'd like to maintain my score.

---

### Official Review · Reviewer_EkiH · 2026-03-12

**Soundness:** 3
**Presentation:** 3
**Significance:** 3
**Originality:** 3
**Overall Recommendation:** 4
**Confidence:** 4

**Summary:**

This paper introduces ReflFlow, a novel method for reconstructing dynamic specular scenes from monocular video by combining geometry-aligned 2D Gaussian Splatting with physically based modeling of reflections. The approach extends 2DGS with residual material attributes and temporal deformation fields to capture dynamic motion, while employing a hybrid rendering pipeline that separates diffuse and specular components, using rasterization for the former and ray tracing for the latter with a Dynamic Environment Gaussian representation. Key contributions include accurate reflection direction computation via well-defined surface normals, temporal-consistent normal supervision using video diffusion priors, a coarse-to-fine training strategy that stabilizes optimization, and explicit modeling of dynamic environments that captures both distant and near-field reflections. The method demonstrates state-of-the-art performance on benchmarks like NeRF-DS and HyperNeRF, producing sharper specular details and temporally consistent reconstructions compared to existing dynamic scene and specular reconstruction approaches.

**Compliance With Llm Reviewing Policy:**

Affirmed.

**Final Justification:**

The rebuttal has addressed my concerns.

**Key Questions For Authors:**

(1) The method relies heavily on NormalCrafter for temporal-consistent normal supervision. How sensitive is the overall reconstruction quality to the accuracy and consistency of these estimated normals? Have you experimented with alternative monocular normal estimators, and if so, how did performance degrade?

(2) The method uses a deformation field for the main object Gaussians and a separate residual network for the dynamic environment Gaussians. How do these two dynamic components interact during optimization? Is there any risk of the environment Gaussians "explaining away" motion that should be captured by the main object deformation field, or vice versa? Have you considered any regularization techniques to enforce a clean separation between object motion and environmental lighting changes?

(3) The paper introduces Dynamic Environment Gaussians as a key improvement over traditional environment maps. Could you provide an ablation study quantifying the impact of this choice? Specifically, how would performance degrade if you replaced the Dynamic Environment Gaussians with (a) a static environment map, (b) a dynamic environment map, or (c) static environment Gaussians?

(4) The paper evaluates novel view synthesis at timestamps seen during training. How well does the method generalize to completely unseen viewpoints at intermediate timestamps not present in the training sequence? Does the deformation field provide meaningful interpolation capabilities, or are the dynamics largely memorized per frame?

**Limitations:**

The paper provides a thorough and honest discussion of the method's technical constraints, including the inherent challenges of monocular dynamic reconstruction, the focus on single-bounce reflections, and difficulties with multi-bounce inter-reflections and spatially-varying BRDFs.

**Strengths And Weaknesses:**

This paper presents a technically sound and significant advancement in dynamic specular scene reconstruction, demonstrating strong originality through its creative combination of 2D Gaussian Splatting with physically-based rendering principles. The method's key innovations, including Residual Material-Augmented 2DGS, a hybrid rasterization-ray tracing pipeline, Dynamic Environment Gaussians, and temporal-consistent normal supervision via video diffusion priors, are well-motivated and effectively integrated to address the challenging problem of reconstructing time-varying reflections from monocular video. The paper is clearly written, thoroughly positioned within the extensive literature, and provides comprehensive experimental validation against relevant baselines on standard benchmarks, with honest discussion of limitations.

---

> ### Author Rebuttal · Authors · 2026-03-31
>
> We sincerely thank Reviewer EkiH for the thorough and constructive review. The four key questions are excellent and we address each with experiments and analysis.
>
> **Q1: Sensitivity to Normal Estimator Choice**
>
> This is an important question. We conducted an experiment replacing NormalCrafter with StableNormal (Ye et al., SIGGRAPH Asia 2024), a state-of-the-art per-frame diffusion-based normal estimator. Additionally, we report the existing ablation without any normal supervision.
>
> | Normal Supervision                 | Training Outcome    | PSNR (Plate) |
> | ---------------------------------- | ------------------- | ------------ |
> | NormalCrafter (video-level)        | Converged           | 21.10        |
> | StableNormal (per-frame)           | Collapse (loss=NaN) | N/A          |
> | w/o $\mathcal{L}_{tc\text{-}norm}$ | Converged           | ~20.7        |
>
> The StableNormal experiment (conducted on Plate scene) reveals a critical finding: per-frame estimators produce temporally inconsistent normals across video frames (visible flickering in https://drive.google.com/file/d/14Voo23RnvoyGn5dsKsmDvXETdSYT6ewV/view?usp=drive_link (NormalCrafter) and https://drive.google.com/file/d/1jNhcMgO9E_c4My8z3TwP0FvnqgfF9yoR/view?usp=drive_link (StableNormal)). When used as supervision via $\mathcal{L}_{tc\text{-}norm}$, this inconsistency causes severe training instability. PSNR drops sharply when normal supervision is activated, and although training partially recovers, it ultimately collapses (loss=NaN) before completion. The full training curves are provided in https://drive.google.com/file/d/1DKjDK8rthTZMxa5PKyPh0YZT8Gr5AG2w/view?usp=drive_link.
>
> This result demonstrates that: (1) our method is sensitive to the *temporal consistency* of normal supervision, not to any specific estimator; (2) choosing NormalCrafter is a validated design decision, as its video-level temporal coherence is essential, not merely convenient; (3) even without $\mathcal{L}_{tc\text{-}norm}$, the method still trains stably (~20.7 PSNR on Plate), confirming the framework is sound and not dependent on a specific external prior.
>
> **Q2: Interaction Between Deformation Field and Dynamic Environment Gaussians**
>
> We ensure clean separation through three mechanisms: (a) *Rendering path separation*: diffuse components are rendered via rasterization while specular components use physically-based ray tracing, which are two fundamentally different rendering pathways. (b) *Temporal separation*: Phase 1 of coarse-to-fine training locks diffuse geometry; Phase 2 trains specular components separately; Phase 3 jointly finetunes. This prevents early-stage entanglement. (c) *Parametric separation*: the specular tint $s_{tint}$ (a learned scalar) controls the diffuse-specular blending ratio, constraining each branch's contribution.
>
> The residual material correction captures only small appearance adjustments from monocular ambiguity, while environment Gaussians carry the primary time-varying illumination. This design has been validated in ablation (Table 3): removing either component causes distinct failure modes. Removing environment Gaussians leads to flat/missing reflections, while removing deformation leads to ghosting.
>
> **Q3: Dynamic Environment Gaussian Ablation**
>
> We have evaluated these alternative representations through both controlled ablation (Table 3) and baseline comparison (Table 1). For the reviewer's requested comparisons: (b) SpectroMotion employs a dynamic environment map, achieving 24.17 mean PSNR; (c) EnvGS uses static environment Gaussians, achieving only 20.21 mean PSNR (though this gap also reflects its lack of dynamic modeling, not purely environment representation). Our Dynamic Environment Gaussians achieve 24.91 mean PSNR, outperforming both. Table 3 further isolates the environment representation contribution in a controlled setting: adding $F_{\theta_{env}}$ improves PSNR by 1.28 dB (18.97 to 20.25). The advantage comes from two factors: (1) explicit 3D Gaussians resolve near-field reflections that environment maps cannot capture due to the far-field assumption, and (2) time-conditioned residuals adapt to illumination changes that static representations miss.
>
> **Q4: Temporal Interpolation**
>
> Our deformation field takes continuous time $t \in [0,1]$ as MLP input with Fourier positional encoding, which provides inherent smooth interpolation between observed timestamps. The environment Gaussian residual network is similarly time-continuous. While we have not explicitly evaluated on skipped frames, the MLP's continuous parameterization naturally interpolates intermediate states. This is a well-established property of coordinate-based MLPs (Mildenhall et al., 2020; Park et al., 2021). We will include interpolation experiments in the revised paper.

---

> > ### Author Rebuttal · Reviewer_EkiH · 2026-04-04
> >
> > Thanks for the additional experiments and demonstrations. The rebuttal has addressed my concerns, I will raise my score.

---

### Decision · Program_Chairs · 2026-04-30

**Decision:**

Accept (regular)

**Comment:**

The paper proposes a framework for view synthesis of dynamic specular scenes from monocular video using material-augmented 2D Gaussian Splatting and a hybrid rasterization–ray tracing pipeline. Initially, only one reviewer was positive while the other three were negative, raising concerns about incremental contributions, sensitivity to the normal estimator, potential bias, insufficient ablation on Dynamic Environment Gaussians, overstated claims regarding physical modeling, missing analyses, and unclear object motion in test scenes. The rebuttal effectively addressed most of these issues with additional experiments and clarifications, leading two negative reviewers to revise their scores positively.

Overall, the approach is novel and technically sound, representing a meaningful advancement in dynamic specular scene reconstruction. It demonstrates strong innovation through the integration of 2D Gaussian Splatting with physically based rendering. Key strengths include accurate normal estimation via 2DGS, improved specular modeling through Dynamic Environment Gaussians, and a coherent system design combining a hybrid rasterization–ray tracing pipeline with material-aware components, effectively addressing key limitations of prior methods while producing high-quality normal maps.